# Yeast Fermentation at Low Temperatures: Adaptation to Changing Environmental Conditions and Formation of Volatile Compounds

**DOI:** 10.3390/molecules26041035

**Published:** 2021-02-16

**Authors:** Wiktoria Liszkowska, Joanna Berlowska

**Affiliations:** Department of Environmental Biotechnology, Lodz University of Technology, Wolczanska 171/173, 90-924 Lodz, Poland; wiktoria.liszkowska@dokt.p.lodz.pl

**Keywords:** yeast, low-temperature fermentation, food production, volatile organic compounds

## Abstract

Yeast plays a key role in the production of fermented foods and beverages, such as bread, wine, and other alcoholic beverages. They are able to produce and release from the fermentation environment large numbers of volatile organic compounds (VOCs). This is the reason for the great interest in the possibility of adapting these microorganisms to fermentation at reduced temperatures. By doing this, it would be possible to obtain better sensory profiles of the final products. It can reduce the addition of artificial flavors and enhancements to food products and influence other important factors of fermented food production. Here, we reviewed the genetic and physiological mechanisms by which yeasts adapt to low temperatures. Next, we discussed the importance of VOCs for the food industry, their biosynthesis, and the most common volatiles in fermented foods and described the beneficial impact of decreased temperature as a factor that contributes to improving the composition of the sensory profiles of fermented foods.

## 1. Introduction

Yeasts are unicellular fungi found commonly in the natural environment, which are also used in various industries [1]. Their fermentative abilities have been used by humans for thousands of years. In the past, fermentation processes were considered mostly an effective preservation technique [2,3]. In the nineteenth century, the developing food industry designed a new generation of fermentation equipment, enabling large-scale fermentation and ensuring conformity to strict safety regulations [4]. The enzymatic activity of yeast now plays an essential role in the production of a wide range of fermented food products (Figure 1). Under anaerobic or oxygen-rich conditions, it is possible to obtain ethanol, carbon dioxide, and various organic compounds through fermentation, all of which have applications in food processing [5]. Alcoholic beverages can be produced with high percentages of ethanol. Bakery products can be leavened with large amounts of carbon dioxide. Fermentation by-products can be used to impart the desired sensory profiles to the final products, contributing to creating specific textures, aromas, and tastes [6,7].

Yeasts are also used in other industries as production organisms for various enzymes and other compounds. Because of their negligible production of undesired secondary metabolites, they are well adapted for use as host organisms and can be applied in a wide range of large-scale biotechnological processes [3,10,11]. For instance, the pharmaceutical industry uses *Saccharomyces cerevisiae* strains to produce human insulin as well as different vaccines against viruses [12]. In the chemical industry, recombinant yeasts, especially *Kluyveromyces pastoris, Saccharomyces cerevisiae*, *Blastobotrys adeninivorans*, *Ogataea polymorpha*, *Kluyveromyces lactis*, *Schizosaccharomyces pombe*, *Yarrowia lipolytica*, *Pseudozyma* spp., and others, are used to produce enzymes on a large industrial scale. Finally, yeasts have been applied in the fuel industry to produce sustainable energy sources from waste biomass [13,14], thanks to their ability to produce and high resistance to bioethanol [15].

Recently, the potential positive effects of fermented products have been the subject of extensive study. Most fermented products have been found to have a positive influence on human health, especially on the skeletal system and internal organs, as well as on the cardiovascular and nervous systems [9,16,17,18]. Fermented foods can be carriers for probiotic microbes and also provide various bioactive compounds that possess prebiotic features and thus contribute to maintaining healthy intestinal microflora. These include *S. cerevisiae*-derived compounds such as β-glucan, glutathione, glutathione disulfide, ornithine, and spermidine [19]. The metabolic pathways of yeasts are well understood, particularly for species such as *Saccharomyces cerevisiae* and *Saccharomyces boulardii*, which are known to possess probiotic features with beneficial effects on human health and may be administrated to patients in the form of dietary supplements or functional food [20]. During fermentation processes conducted by yeast, it is possible to obtain numerous organoleptic bioactive volatiles and non-volatile compounds, such as peptides, amino acids, vitamins, minerals, and polyphenols [21]. Fermented products, especially dairy and grain-derived products, are good sources of vitamins such as vitamin B_2_, vitamin B_9_, and vitamin K [22,23]. Recently, the impact of melatonin on fermentation processes has attracted increasing attention. In general, this yeast-derived bioactive compound (from both *Saccharomyces* and non-*Saccharomyces* yeasts) has been found to have a beneficial effect against oxidative and UV stress [24,25,26].

Various bacteria also have fermentation activity and can be used in fermentation processes with yeast [27]. Bacteria can increase the bioavailability of various compounds naturally present in the raw materials subjected to fermentation [28]. For instance, compounds with low bioavailability can be freed from cells by enzymatic degradation of the cell wall, thus facilitating absorption [29,30]. Such cooperation enables the production of fermented products enriched with different compounds (including volatiles). There are numerous works in the literature concerning the relations between microorganisms, such as certain bacteria or non-*Saccharomyces* yeasts, with enzymatic activities different from those of *S. cerevisiae* and the ability to catalyze the synthesis and release reactions of various volatile organic compounds [31]. One study, performed by Tran and co-workers, presented the relationship between acetic acid bacteria (AAB) and yeasts during fermentation of kombucha. They showed that AAB stimulate *S. cerevisiae* yeast to produce metabolites under oxygen-limited conditions [32]. Another study, conducted by Tufariello and co-workers, was devoted to the improvement of the sensory profile of Negroamaro red wine at both the laboratory and the industrial scale using an innovative mixed starter formulation (*Saccharomyces cerevisiae* NP103, *Candida zemplinina* 35NC1, and *Lactiplantibacillus plantarum* LP44). The combination of yeasts and bacteria led to increased concentrations of key compounds, including esters, which are mainly responsible for the characteristic fruity and floral aroma of wine [33]. Nevertheless, yeast–yeast relations are also important in modulating the chemistry of edible matrices. Particularly interesting co-cultures include *Saccharomyces* and non-*Saccharomyces* species. Non-*Saccharomyces* yeasts are considered to have several beneficial effects, including limiting or enhancing target post-fermentation metabolites present in the final product (for example, to reduce alcohol content). They also have the ability to maintain a good fermentation rate in environments with high sugar concentrations [34]. An interesting study was performed by Englezos and co-workers on the non-*Saccharomyces* species *Starmerella bacillaris* as a potential co-culture yeast for mixed cultures with *S. cerevisiae*. The mixed cultures showed the ability to increase concentrations of key volatile compounds in wine compared to pure *S. carevisiae* [35].

Fermented dairy products, especially kefir, are a rich source of bioactive peptides with antibacterial, antioxidant, and even antihypertensive features, which have a beneficial impact on human health [36,37,38]. Polyphenols are another significant group of compounds in fermented products, particularly in alcoholic beverages, and are considered beneficial for humans. For instance, in their extensive review, Adebo and Medina-Meza gathered data from various studies on the influence of fermentation on phenolic compounds in whole grains. The collected results showed increased levels of phenols in most cases, with a slight decrease in a few products. This could be related to the unique compositions of the raw material or the specific activity of the microorganisms used [28]. Another interesting review by Annunziata and co-workers examined the literature on bioavailable fermentation-derived compounds, especially short-chain fatty acids. The general conclusion was that these compounds are formed in large amounts during the fermentation process. Their bioaccessibility was also high due to enzymatic activity during fermentation, which led complex structures to degrade into smaller particles better absorbed in the intestinal environment [39]. Numerous studies have shown that polyphenols possess strong antioxidant properties [40,41,42]. Yeasts may be involved in determining the final compositions of these compounds in different fermented products [28].

Very important compounds from the perspective of the food industry are volatile organic compounds (VOCs), which have a key role in determining the sensory profiles of the final food products [43]. Volatile organic compounds are a group of volatile organic molecules at room temperature with a molecular weight of up to 300 Dalton [43]. They are commonly found in fermented foods. During fermentation, huge amounts of these compounds are synthesized or released from raw materials. For instance, organic acids also have significant preservative properties against food spoilage [44]. Their activity depends on the pH and undissociated form. The organic acids with the best antimicrobial activity are acetic acid, lactic acid, propionic acid, benzoic acid, and sorbic acid [45]. Due to the importance of VOCs in food processing, particularly those derived from yeast fermentation, there has been extensive research in this area. The biodiversity of fermentative activity of *Saccharomyces* stains and the metabolic abilities of non-conventional yeast is a huge research area [46]. An extensive study was performed by Berbegal and co-workers, who studied two non-*Saccharomyces* strains (belonging to the species *Metschnikowia pulcherrima* and *Torulaspora delbrueckii*) together with *Saccharomyces* yeasts. They found a correlation between the increased number of yeast cells in the fermentation environment and the increased amount of VOCs, for bot single strains and mixed cultures containing different strains [31]. These compounds will be discussed in detail in Section 3.

This review focuses on the adaptation of yeast to low-temperature processes, in particular the food industry. Due to the relatively large number of microorganisms used in food production, there have been many studies on how microorganisms adapt to reduced temperature processes, which have a number of advantages in terms of both the quality of the final products and the optimization of the processes themselves, including lower cost. In particular, we will focus on the role of yeasts in the formation of VOCs at lower temperatures. During fermentation, VOCs are synthesized and released into the environment. Apart from their preservative properties, which increase the value of the product itself, VOCs have a key role in ensuring that the final product has an acceptable sensory profile for consumers. Without VOCs, it would not be possible to obtain the desired flavors and aromas of products such as alcoholic beverages and bread. We therefore examine the ability of yeast to adapt to low-temperature environments and the effect of temperature on their ability to increase the levels of VOCs in fermented foods. We will also describe the characteristic sensory profiles of various products of yeast fermentation at different temperatures.

## 2. Conducting Biotechnological Processes at Low Temperature

The implementation of various biotechnological processes for industrial purposes is known as white biotechnology. This name is understood to highlight the environmental benefits of using microorganisms instead of chemical reactions. The main purpose is to develop less harmful chemical-free technologies, using natural materials as substrates, which should be converted with the highest-possible efficiency in terms of energy and waste [47]. Such processes have replaced numerous large-scale chemical reactions, contributing to reducing harm to the environment [48,49]. White biotechnology typically employs post-production residues as substrates for other processes, or renewable resources such as sugars and materials of plant origin [50]. Using microorganisms (bacteria, yeasts, algae, fungi, and many others), it is possible to convert substrates into a wide range of products, including chemicals, pharmaceuticals, food colorants, vitamins, and food additives, as well as bioplastics and biofuels [51]. Bioprocesses have the following advantages:increased reaction ratehigher conversion efficiencyproducts with better puritydecreased energy consumption throughout the processreduced generation of chemical-derived waste [47]

Recently, biotechnological processes have been optimized to operate at lower temperatures. However, it was found that the enzymatic activity of mesophilic enzymes decreased at lower temperatures, resulting in a longer process. A review by Margesin and Schinner devoted exactly to this topic provides much useful information regarding the growth rate of psychrophilic, psychrotrophic, and mesophilic microorganisms. As expected, mesophiles grow slower at lower temperature but faster as the temperature increases, unlike psychrophiles and psychrotrophs. Temperature may also affect the enzymatic activity of mesophiles. In microorganisms adapted to low temperature, this obstacle is overcome by the formation of more enzymes, which compensates for the slow rate of enzymatic activity. Furthermore, the biosynthesis of polysaccharides in mesophiles is lower at low temperatures, so they have developed genetic mechanisms that allow them to accumulate more sugars (for instance, trehalose) [52].

Although mesophilic and psychrotrophic microorganisms are capable of producing enzymes at low temperature, enzymes from psychrophiles conduct reactions more efficiently in the temperature range of 0–30 °C. Psychrophilic and psychrotrophic yeast and yeast-derived cold-active enzymes can adjust to different low-temperature fermentation processes [53,54]. Thus, they can be applied to industrial bioprocesses conducted at low temperatures [55,56]. Cold-adapted enzymes such as amylases, invertases, lipases, proteases, and cellulases are used as biocatalysts in sectors including the fuel and food industries, while many psychrozymes are applied in detergent manufacturing [53,57,58]. As a consequence, the focus has shifted to using psychrophilic and psychrotrophic organisms that are better adapted to low temperatures and can produce psychrozymes [59,60]. These enzymes possess the highest conversion efficiency at 10–15 °C and have been applied for industrial purposes [61]. Their most important advantages include:reduced production, storage, and transportation costs due to lower temperature requirements;lower energy use, with no need to heat bioreactors;fewer undesirable by-products due to the lower rate of formation (the substrate is converted to the main product with higher efficiency);reduced losses resulting from the degradation of thermolabile substrates and reaction products;reduced formation of mesophilic microorganisms, which can lead to food spoilage; andthe ability to grow with good yields at temperatures below optimum [60,62,63,64,65].

Various microorganisms undergo environmental stress caused by low temperatures. For instance, probiotic bacteria have developed an interesting response mechanism to culture storage before inoculation or when the final product is stored. Low temperature can have different effects within the cell, reducing enzyme activity during RNA transcription and protein synthesis, for example, or stiffening cellular membranes. To survive under such conditions, probiotics have adapted several mechanisms of response. The most important of these is the induction of *cps* genes (commonly present in lactobacilli). The transcription of *cps* genes leads to the production of cold-shock proteins (CSPs), which are responsible for maintaining active growth conditions for cells [66]. Other proteins that are believed to be involved in the cold stress response in lactobacilli species are AddB, UvrC, RecA, and DnaJ [67].

Given these advantages, there is great interest in developing methods for the industrial-scale production of microorganisms, especially yeast and fungi, with comparable performance to their mesophilic equivalents in fermentation processes conducted at room temperature and below [59].

### 2.1. Yeast Stains Active at Low Temperatures

Psychrophilic and psychrotrophic (cold-adapted) microorganisms are distinguished from mesophiles by their ability to grow at low temperatures [52]. Psychrophilic microorganisms have a maximum temperature for growth of 20 °C or below and are restricted to permanently cold habitats, whereas psychrotrophic microorganisms have maximum temperatures for growth of more than 20 °C. Growth at low temperatures is often associated with thermolability [68]. Such microorganisms can have slower metabolic rates and higher catalytic efficiencies than mesophiles, making them considerably interesting for biotechnological applications [69]. Some cryotolerant strains with good adaptation to low temperature belonging to *Saccharomyces* species (*Saccharomyces uvarum*, *Saccharomyces kudriavzevii*, and *Saccharomyces eubayanus*) can be used in industrial fermentation processes, especially in wine production [70,71,72]. However, most of the studied non-*Saccharomyces* (except for *K. marxianus*) show a lower optimum temperature than *S. cerevisiae* [73].

### 2.2. Acclimation of Yeast to Low Temperatures

Microorganisms have the ability to adapt to stress in response to challenging conditions in both their natural habitat and industrial applications. In the food industry, these stresses are caused mainly by the levels of acidity and alcohol, which can inhibit the growth of microorganisms under certain conditions. However, temperature also determines the growth rate [57,74,75]. Although the mechanisms of the cold stress response are quite well understood, these adaptations in yeast cells are worth describing due to their potential applications in low-temperature food processing.

*Saccharomyces cerevisiae* strains are widespread in both natural and artificial environments, where they are exposed to various temperatures. Their adaptation to subnormal thermal conditions is closely related to genetic changes, which also cause physiological variations [76,77,78]. Although optimal temperatures for *S. cerevisiae* yeasts are around 25–30 °C, they can be successfully stored under both industrial and laboratory conditions at temperatures of around 4 °C, maintaining viability for a long time [79]. Numerous experiments have been performed in which yeasts were subjected to lower temperatures than optimal, mainly through cold shock, and their cellular responses were analyzed. It was concluded that within the yeast genome, there are genes responsible for adaptation to cold. Gene expression is crucial also for the set of proteins that create the proteome during translation (elongation conducted by tRNA). Translation strongly depends on the presence of amino acids (specifically, aminoacyl-tRNA). Without aminoacyl-tRNA, translation is inhibited due to the fact that there is no substrate that can be added to the elongated protein chain. Amino-acid-derived fusel alcohols also have the ability to inhibit the process of protein formation [80]. During changes in the yeast proteome at low temperature, heat-shock proteins (HSPs) play an important role in cell maintenance. They contribute to ensuring the correct folding of polypeptides produced during translation. Under stress conditions, they prevent protein unfolding or help partially unfolded proteins to fold again. An interesting study was conducted by Ko and co-workers (2017), in which carrot-derived *HSP17.7* were inserted into the *S. cerevisiae* genome. This modification was found to increase the viability and growth rate of the yeast cells under stress conditions, such as low temperature. Although its subject was a genetically engineered mutant, this study highlighted the importance of heat-shock proteins [80].

Numerous genes related to cold-response proteins have been identified in *S. cerevisiae* yeast cells (Table 1).

There have been numerous studies on the cellular changes that occur under suboptimal temperature conditions. For instance, Brown and co-workers analyzed the impact of quick temperature changes on yeast. When the yeast was subjected to cold shock, the dynamic changes in temperature triggered genetic mechanisms that led to rapid adaptation to stress. However, under longer exposure to milder but also stressful temperature changes, the yeast was able to almost fully acclimatize by physiological changes [87]. Population genomic studies on *Saccharomyces* yeast revealed unique changes to the original phenotypic characteristics [75,88]. Small genetic changes are considered to be the result of single-nucleotide polymorphisms, frame shift mutations, or possibly insertions and deletions within the genetic code. These variations can lead to alterations in gene expression, slight changes in the structure of molecules, or changes in their functions. Even when they undergo large changes, including changes in the number of gene copies or rearrangements on the chromosomal level (such as segmental translocations, inversions, or duplications), yeasts are able to maintain relatively good cell condition. Other processes that are also known to affect the resistance of yeast to unfavorable conditions include a specific kind of hybridization of two or more microorganisms and so-called introgression, whereby some genes from a given species are introduced into the genome of another organism [75].

To survive in low-temperature environments, yeast has to assimilate a carbon source, which gives it the energy to conduct essential processes. To this end, yeast produces various extracellular hydrolytic enzymes, which have the ability to hydrolyze complex sugars to a form that can be assimilated. This is connected to the assimilation of trehalose, a sugar that accumulates in yeast cells, by inducing the genes TPS1 and TPS2. Synthesized trehalose acts as a carbohydrate reserve and can be utilized by the cells as a source of energy. Under the influence of thermal shock, such as low temperature, significant amounts of trehalose are accumulated by yeast cells [89,90]. Trehalose protects proteins and membrane lipids, stabilizing the cell and maintaining its structural integrity. Significant rises in the levels of this sugar also increase the osmolarity of the cells, thereby improving the mechanical resistance of the cell wall and reducing their sensitivity to lytic enzymes [91].

Under conditions of stress, including low temperatures, genetic changes occur in yeast cells, which, in turn, cause physiological and morphological changes. As well as the fundamental mechanism of trehalose accumulation, *S. cerevisiae* yeast have the ability to assimilate various other compounds, such as proline, and heat-shock proteins responsible for cell maintenance. However, these mechanisms vary, depending on the physiological state of the cell before it is exposed to stress conditions [92]. The most important morphological change is in the structure of cell membranes. Singer and Nicolson proposed the first model for membrane fluidity in 1972, describing a liquid-crystalline lipid bilayer in which proteins are submerged [93]. The cell membrane has several very important functions in microorganisms [94,95]. It is the primary defensive structure, separating the extracellular environment from the inner cell structures. Membranes contain numerous signal proteins, which sense each change in the environment and send information via signal transduction pathways to regulate the expression of specific genes. The composition and modification of lipids present in the membrane also plays an important role in the expression of genes responsible for adaptation to stressful environmental changes [96]. The physiological state of lipids is essential for the proper functioning of all proteins present in the membrane, including sensing proteins such as kinases and ion channels [92,97]. It is possible to manipulate the composition of membrane lipids. Lower membrane fluidity has been found to enable cells to adapt to low temperatures. The most common changes are related to reduced saturation of fatty acids and shorter fatty acid chain lengths [98]. Sterols present in the membrane influence the fluidity of the structure. In general, sterols stabilize and strengthen the cell membrane. They also determine whether lipids are in a crystalline or a more liquid state [82,99].

In the global context of stress response changes and the variability of microorganisms, it is worth mentioning the biodiversity that led to the creation of *Saccharomyces pastrianus*, a yeast species used in the brewery industry. Although classified as a distinct strain, *Saccharomyces pastrianus* shows great similarity to *S. cerevisiae* and *S. eubayanus* (which is a cryotolerant strain). Such hybrids were created in response to cold-brewing temperatures and can be understood as an evolutionary adaptation that helps yeasts to overcome stressful conditions in the fermentation environment [100].

Different physiological changes ensure the viability of cells and their ability to conduct regular enzymatic reactions. At temperatures below optimal for conducting bioprocesses (around 10–20 °C), the growth of yeast slows and its metabolic activity decreases [101]. However, in the food industry, some fermentation processes involving yeast are carried out at lower-than-optimal temperatures, such as fermentation of lager beer and wine making. As already mentioned, lower-temperature processes enable economic savings [102]. To avoid changes in the sensory profile and the loss of nutritional properties, while at the same time preventing contamination by undesired microflora, food processing of fermented food is moving toward adapting processes to milder conditions. To this end, many other temperature-resistant microorganisms are being tested and applied for industrial applications [53]. Temperature strongly affects the viability and activity of microorganisms. To assist with monitoring the conditions of biochemical changes, temperature-based computer programs have been developed based on modeling techniques [103,104].

## 3. Sensory Profile: Impact of Volatile Organic Compounds on the Final Product

The sensory profile of fermented foods and beverages is a complex feature, combining flavor, aroma, appearance, and, in the case of food, texture. Together, these components create the unique characteristics of a given product and determine its quality. Whenever a company introduces a new product to the market, these characteristics are assessed by a food sensory evaluation panel [105].

Volatile organic compounds (VOCs) are low-molecular primary and secondary metabolites produced by microorganisms. In the course of yeast fermentation, they impart characteristic features to food and beverages. In addition to the two main yeast-derived metabolites, ethyl alcohol and carbon dioxide, smaller amounts of VOCs are formed [34]. They possess low polarity and high vapor pressure. In the natural environment, they act as chemical signals responsible for communication between cells [106]. This feature predestines VOCs to be used as biosensors [107]. In different industrial branches, they can be obtained not only by using microorganisms but also via chemical reactions. In industrial applications, especially food manufacturing, they contribute to the characteristic sensory features of the final product [108]. It is worth underlining that under the conditions of fermentation, including the amounts of substrates, additives, and microorganisms, VOCs have a strong influence on the composition of the final product. For instance, Aljewicz and co-workers found that the final concentration of certain aldehydes and ketones in yoghurt increased when β-glucans (isolated from bacteria) were added [109]. Volatile organic compounds can be produced in food by chemical reactions that occur during food processing, such as Maillard reactions [110]. They can also be released from non-volatile precursors in foods and as a result of microbial metabolism [111]. Some volatiles are already present in plant-derived materials that are then subjected to microbial fermentation [34]. Others are formed by the activity of microorganisms with the ability to perform reactions, resulting in the synthesis or release of desired compounds [112]. For instance, volatiles present in bread are derived from three main pathways: lipid oxidation of flour, Maillard reactions, and fermentation conducted by yeasts. However, some VOCs are considered harmful. One, ethyl carbamate, can be found in red wine kept under deficient aging and storage conditions [113]. To ensure high quality and food safety and to avoid the formation of undesirable side compounds, process conditions should be strictly controlled and the microorganisms used should have Generally Recognized as Safe (GRAS) status or be accepted by the European Food Safety Authority (EFSA) [114,115].

Volatiles can also be created by yeast fermentation, mainly during wine, beer, and bread making [116]. Research carried out by Makhoul and co-workers focused on the production of VOCs during fermentation with the use of three commercial *S. cerevisiae* strains. They identified more than 40 VOCs, including alkenes, esters, aldehydes, ketones, alcohols, carboxylic acids, and small amounts of furan derivatives and sulfur compounds. These are mainly derived from yeast metabolism and are necessary in order to obtain the desired sensory profile of the final product. Some of them are released during the baking process [117]. An additional study by Makhoul and co-workers revealed the role of two *S. cerevisiae* strains in creating the specific sensory profile of baked bread. In all breads, the aroma profiles comprised the main VOCs produced by the yeasts, in different amounts depending on the strain used [118].

Numerous VOCs are formed during food processing. These can be divided into the following groups: higher alcohols, organic acids, fatty acids, aldehydes, ketones, esters, and volatile phenols [119]. Higher alcohols, produced via the Ehrlich pathway, are synthesized by yeast through the anabolic glucose pathway but also in a catabolic reaction from certain amino acids. In both cases, α-ketoacids are the key intermediates. For instance, leucine is converted to isoamyl alcohol, valine to isobutyl alcohol, and phenylalanine to 2-phenylethanol [120]. Due to the variety of yeasts used in food production and the composition of the substrates, different amounts of higher alcohols are present in fermented foods and drinks [121,122,123]. Organic acids are another group of VOCs [45]. Organic acids possess preservative acidic properties and can be found naturally in food products. They may also be created during various chemical reactions, especially microbial activity. They contribute to natural food preservation, which is the reason for the long shelf lives of fermented foods and beverages [124]. In their review of the research, Kourkoutas and Proestos identified organic acids, among other compounds, as contributing to preservation against food spoilage [125]. The most frequently occurring organic acids that act as preservatives and possess antimicrobial action are acetic acid, benzoic acid, citric acid, formic acid, lactic acid, propionic acid, and sorbic acid. These acids inhibit the growth of undesirable microflora by decreasing the pH of their intracellular environment. This prevents the microflora from performing basic molecular processes [116,126,127]. In general, microbial-derived organic acids are synthesized through the citric acid cycle (TCA) or via the reduction of citric acid by various enzymes. The main secondary metabolite of yeast fermentation is succinic acid. Smaller amounts of intermediates, such as pyruvic, malic, fumaric, oxaloacetic, citric, α-ketoglutaric, glutamic, propionic, lactic, and acetic acids, are detectable in the post-fermentation environment [128]. Short-chain fatty acids (SCFAs), which are small organic monocarboxylic acids that can be treated as a subgroup of organic acids, are produced by microorganisms as end products of fermentation. The most common SCFAs are acetic, propionic, and butyric acids. One of the methods for obtaining these compounds is hydrolysis of acetyl-CoA, propionyl-CoA, and butyryl-CoA, which are derived from the reduction of pentoses and hexoses to the desired SCFAs [129]. Another significant group of VOCs is composed of carbonyl compounds, aldehydes and ketones. These are formed enzymatically by the reduction of carboxylic acids or non-enzymatically as a result of oxidation-related alcohol. They may also be formed in reactions between certain amino acids and α-dicarbonyls or as a result of the condensation of activated fatty acids due to lipoxygenase activity. The best-documented yeast-derived volatile is acetaldehyde, which can be also be obtained by the decarboxylation of pyruvate [130,131,132]. The sensory perception of acetaldehyde changes depending on the amount in the final product. A low concentration enhances fruitiness, whereas increasing levels give a nutty aroma and then a smell like rotten apples [130].

Esters are an extremely large group of compounds that are mainly responsible for the aroma of food products. In general, except for substrate-derived esters, they are formed through yeast metabolism and then are released to the fermentation environment. The best-studied group is acetate esters, which are the most abundant esters in fermented foods and beverages. Acetate esters are formed during the reaction of acetyl-CoA with higher alcohols, whereas fatty acid ethyl esters are formed due to the activity of yeast enzymes and acetyl-CoA ethanolysis. Acetyl-CoA is produced during the synthesis or degradation of fatty acids and carbohydrates [133]. The importance of these compounds is significant due to specific fruity aromas, which give characteristic sensory features and determine the quality of wines and top-fermenting beers, in particular [134].

The final significant group of VOCs present in fermented products comprises volatile phenols. These are produced (among other pathways) by the decarboxylation of hydroxycinnamic acids (HCAs), which are the C3–C6 secondary metabolites of yeast. Many yeast species are capable of performing reactions leading to the creation of phenols, including commercial *Saccharomyces* yeasts and wild-type *Saccharomyces*, *Kloeckera*, *Rhodotorula*, *Cryptococcus*, *Candida*, and *Pichia*. These reactions result in the formation of 4-vinylphenols (4VPs), such as 4-vinylguaiacol by the decarboxylation of ferulic acid and 4-ethylguaiacol by the reduction of 4-vinylguaiacol, which is responsible for phenolic off-flavors in fermented foods [135,136,137,138]. Miyagusuku-Cruzado and co-workers found that some lactic acid bacteria also have the ability to decarboxylate HCAs. This could be useful when designing the fermentation process and selecting appropriate strains of bacteria or yeast, depending on the desired sensory profiles [139]. The most common VOCs in fermented products are presented in Table 2, which shows the variety of volatiles in foods processed with microorganisms.

### 3.1. Food Processing Conducted with Yeast at Low Temperatures

Mesophilic yeast has been adapted to perform low-temperature fermentation in the food industry. The activity of mesophilic yeast is optimal at 20–45 °C, and processes conducted at lower temperatures result in prolonged time of fermentation. However, in food manufacturing, certain yeast species are commonly used in low-temperature fermentation (Table 3).

An example of a strain that grows well at lower temperatures is *Saccharomyces pastorianus*. This species is used in the production of lager beer fermented at 10–12 °C [42,104,140]. As well as ethanol and carbon dioxide, various sensory compounds are formed during the process. These are derived from the substrates, barley malt and hops, but may also be metabolized by the microorganisms or result from inappropriate storage conditions [42,147,148]. The most important volatile compounds formed during yeast fermentation are esters. Esters give a characteristic floral and fruity aroma. Yeast cells synthesize esters intracellularly, which then diffuse through cell membranes. In the case of lager yeast, the majority of aroma-active esters can be found both inside cells and in the fermenting medium. This is the reason why ale beers are richer in specific aromas than lagers [149,150]. Higher alcohols are another significant group of flavor compounds produced by yeasts. The most abundant higher alcohols with the highest quantity are amyl alcohol and isobutyl alcohol [151]. These alcohols strongly affect the drinkability of beer. The more of these alcohols are present in beer, the heavier the beer [152]. Carbonyl compounds are a third, relatively small group of yeast-derived by-products. They also have an impact on beer drinkability. The most common is diacetyl (2,3-butanedione), which acts as a specific indicator of fermentation and beer maturation quality. Large amounts of this compound result in undesirable aroma and flavor. In the case of lager beers, it is important to remove diacetyl from the maturing beer or to limit its synthesis. On the other hand, when the temperature of beer maturation is relatively low, the rate of diacetyl formation is negligible [153,154].

Wine fermentation requires relatively low temperatures compared to other food bioprocesses. The whole process lasts from a few days up to two months and imparts to the final product a complex taste and aroma. In the case of white and rosé wines, low-temperature fermentation provides greater flavor complexity [77,155]. Given that lower temperature results in the retention of more volatile compounds in wine, numerous studies have examined the changes in VOC composition and the influence of lower temperatures on production strains [156].

Samoticha and co-workers investigated the effect of temperature (12 and 20 °C) on the production of VOCs by different yeast strains (*S. cerevisiae* and *S. bayanus*) during white wine fermentation. Although the temperature during fermentation did not significantly affect the composition of VOCs, the yeast strain and temperature during maturation had a great effect on the formation of the main volatiles. Esters, as expected, were the largest group of VOCs, both in fresh and mature wine. The largest amounts of esters were produced by *S. cerevisiae* yeast at 12 °C. Other amounts of volatiles were also higher at lower temperatures. These results suggest that temperature is not as significant as the yeast strain used for wine production but that lower-temperature processes lead to better sensory profiles than those conducted at higher temperatures [157].

Another very interesting study focusing on the isolation and adaptation of yeasts to low-temperature food processes was performed by Kanellaki and co-workers. They isolated two cryotolerant *S. cerevisiae* strains (YM-84 and YM-126), immobilized them on different supports, and monitored the fermentation rates at different temperatures (0, 5, 7, 13, and 27 °C). At 13 °C, wine was produced with a good ethanol content. The concentrations of key volatiles, such as higher alcohols, were lower than expected; however, the levels of compounds including ethyl acetate increased. Overall, the authors concluded that low-temperature food processing methods show potential for industrial applications due to possible aroma and taste enhancement [158]. A similar conclusion was drawn by Kregiel and co-workers. They investigated the ability of producing given VOCs by yeast at 10 °C (*S. pastorianus* yeast used in the production of lager beer) immobilized on hydroxylapatite and chamotte. Yeasts immobilized on chamotte produced higher amounts of VOCs than those on hydroxylapatite. Although these values were lower than those obtained from free cells, they show potential for industrial applications [159].

Torija and co-workers studied the influence of different temperatures on fermentation kinetics of *S. cerevisiae* and the composition of basic volatile compounds in final products. The temperature range was set to 15–35 °C. The results showed that lower temperatures prolonged the stationary phase of yeast growth (it lasted until the end of fermentation), which may have been a result of reduced accumulation of intracellular ethanol, high levels of which are toxic to yeast cells. It was observed that as the temperature increased, the final amount of alcohol decreased due to the lower ethanol yield. This may have been caused by the fact that the substrate was used not only for ethanol production but also for other by-products. Consequently, the sum of by-products decreased with the temperature. The final amount of carbon dioxide also increased at lower temperature (15 °C). It may be concluded that via its effect on yeast growth and metabolism, temperature also affects the composition of the final wine product [156].

Another interesting study related to the production of VOCs by yeast during wine fermentation at different temperatures was carried out by Molina and co-workers. They compared wine fermentation under anaerobic conditions at 15 °C and 28 °C using *Saccharomyces cerevisiae*. An artificial fermentation medium consisting of glucose, fructose, and amino acids was used to a certain which VOCs were produced directly by the yeast, without grape precursors. The fermentation rate was 2.5 times faster at 28 °C, and as expected, the final amount of ethanol was higher when the temperature was lower. In contrast, glycerol content increased at higher temperature. Most of the VOCs were synthesized within the exponential growth phase; however, at 15 °C, the overall concentration of volatiles was significantly higher. Although this study revealed better aroma properties at lower temperatures, it did not examine normal industrial conditions. Nevertheless, it is possible to obtain such conditions using suitable yeast strains [160].

Not only free yeast but also immobilized yeast has been studied to examine the influence of fermentation temperature on the efficiency of VOC production [161]. Bakoyianis and co-workers analyzed changes in the amounts of volatile compounds during fermentation at temperatures from 7 °C to 27 °C. They used immobilized yeasts on γ-alumina, kissiris, and alginates, with free cells as a reference. They concluded that yeast immobilized on an inorganic support produced lower amounts of volatiles than yeast immobilized on organic alginates. The immobilized cells also produced lower amounts of higher alcohols and ethyl acetate compared to free cells. The final product of low-temperature fermentation using immobilized yeast was also considered to have improved aroma and taste [162]. Another study conducted by Bakoyianis investigated low-temperature wine making by yeast immobilized on gluten pellets, with similar conclusions [163].

In comparison to wine and beer making, where lower-temperature processes are easier to adopt, using low temperatures for traditional bread making is much more difficult. However, introducing new conditions for fermentation or a new yeast strain instead of classic baker’s yeast could enable the optimization of existing bioprocesses and increase the quality of final products by retaining more volatile aroma compounds and improving the structure of the bread itself. Several studies have been conducted in which the temperature was lowered during the fermentation process.

An interesting study on the effects of fermentation temperature and yeast concentration on the aroma profile of bread was performed by Birch and co-workers. Fermentation was carried out at three different temperatures (5, 15, and 35 °C) and with three different concentrations of pressed baker’s yeast *S. cerevisiae* (20, 40, and 60 g/kg flour). The key parameters of the experiment are presented in Table 4.

Samples of crumb from the final products were subjected to quantitative analysis. In total, 45 aroma compounds were identified that were derived mainly from yeast metabolism, and their quantities increased with the concentration of yeast. On the other hand, decreasing the temperature to 5 °C increased the formation of three main esters (ethyl acetate, ethyl hexanoate, ethyl octanoate). Lowering the temperature to 5 °C with a simultaneous increase in the concentration of yeast to 60 g/kg flour was found to provide a high-quality final product with large amounts of esters [112].

A similar study regarding bread crust was performed by Nor Qhairul Izzreen and co-workers. The temperature variants were 8, 16, and 32 °C, with different concentrations of yeast (2%, 4%, and 4%) of flour. Fresh baker’s yeast (*S*. *cerevisiae*) was also used as a fermentation microorganism. The key parameters of the experiment are presented in Table 5.

The levels of volatile compounds in samples of the bread crust were analyzed. In all, 28 volatiles were confirmed, with 30 others identified tentatively. Bread samples fermented at 8 and 16 °C resulted in increased amounts of three esters (ethyl acetate, ethyl octanoate, and ethyl hexanoate) [163]. This is compatible with results reported by Birch and co-workers. Although Birch and co-workers found in the case of bread crumb that the most promising results were obtained with a higher concentration of yeast fermenting at lower temperature, Nor and co-workers discovered that for crust, a higher amount of yeast fermented at 32 °C gave better results. Nonetheless, the various similarities between these two studies may point the way for the development of good fermentation conditions at lower temperature.

A summary of all the cited studies is presented in Table 6

Based on the literature, there is, therefore, substantial potential for the development of efficient fermentation methods that could retain more of the volatiles responsible for the aroma, taste, and other characteristics of final products. Such methods could use not only *Saccharomyces* strains but also other yeasts with similar fermentation activity.

### 3.2. A Brief Study of Alternative Yeasts Used in Bread Making

Recently, studies have considered other yeast strains, not just *Saccharomyces cerevisiae*, with the ability to produce non-toxic by-products and large amounts of carbon dioxide. One such strain is *Pichia anomala* [164,165]. This microorganism is widespread in nature and is known to possess good stress tolerance (including tolerance to high concentrations of ethanol). Due to its microbial properties, it can be used for the preservation of food and pharmaceuticals [166]. During fermentation, it may contribute to flavor enhancement. *Pichia anomala* is a particularly good yeast species for sourdough fermentation [167]. Mo and Sung compared the production of white pan bread using *S. cerevisiae* and *P. anomala* under standard conditions of dough fermentation. Analysis revealed that the composition of volatile flavor compounds in bread fermented by *P. anomala* was better than that in bread fermented by *S. cerevisiae* (53 flavor compounds were detected in comparison to 38, respectively). In sensory analysis, the *P. anomala*-fermented bread was also rated as having slightly higher overall acceptability [168].

Other interesting yeast strains that can be used for sourdough fermentation belong to the species *Candida*. A study on rye bread dough by Häggman and Salovaara revealed that *Candida milleri* has good fermenting properties, with an excellent rate of carbon dioxide production in comparison to classic baker’s yeast, almost independently of fermentation temperature. They also compared the leavening capacity (mL CO_2_/100 g dough) of *C. milleri* and classic baker’s yeast. The leavening capacity of *C. milleri* was approximately double that for baker’s yeast at all of the studied temperatures (22, 25, 28, and 31 °C) [169,170].

The species *Kluyveromyces* has been investigated as a possible alternative to classic baker’s yeast. A particularly important strain for the food industry is *K. lactis*, which was the first non-conventional species after *S. cerevisiae* to be given the status of Generally Regarded as Safe (GRAS) [171]. In contrast to *S. cerevisiae*, which do not possess such activity, *K. lactis* has the ability to use lactose as a carbon source. Its use as a sourdough starter culture may be expected, therefore, to be very effective [172]. Caballero and co-workers prepared bread with whey or lactose, which was fermented by a close strain, *Kluyveromyces marxianus*, or by *S. cerevisiae*. In lactose-rich dough, *K. marxianus* showed higher proofing activity than classic baker’s yeast. The aroma profile of the final product was also better. Therefore, it was concluded that *K.*
*marxianus* is a good alternative for the production of sourdough bread [173]. A similar study was performed by Akyüz and Mazı. Based on the results, it was concluded that even if the bread fermented by *K. lactis* received a slightly worse sensory evaluation, this strain should be considered as an alternative to baker’s yeast for the production of whey-/lactose-enriched bread [174,175]. The same conclusion was drawn one year later, when Mazı carried out a similar study [174,175].

## 4. Conclusions

The aim of this review was to present the potential of yeasts for use in food manufacturing conducted at temperatures lower than those used currently. Many genetic and physiological adaptation mechanisms were presented. Furthermore, studies cited by us are quite positive in the context of implementing lowered temperature to existing bioprocesses in food industry. We also wanted to show the impact of yeasts on fermented products, especially on volatile organic compound formation. Many studies have proved the dependence between fermentation conducted by these microorganisms and the formation characteristic volatiles. Furthermore, it is confirmed that low temperature provides a better sensory profile of fermented foods. Although *Saccharomyces* strains are still the main yeasts used in food fermentation processes, numerous studies have shown the potential of other organisms, such as *Pichia* or *Kluyveromyces* strains. Based on the results of the literature discussed here, it is possible for yeast to adapt to low-temperature fermentation processes. Thanks to that ability, it should be possible to optimize the production of food products such as bread by improving the composition of the sensory profile of the final product and reducing the costs of the entire process due to the introduction of new industrial conditions. Despite the potential of yeasts to perform fermentation under unusual conditions, the whole process must be managed in such a way as to maintain fermentation efficiency compared to standard methods, while also avoiding the potential risk of toxic metabolites produced by yeasts. More research to this end is required.

## Figures and Tables

**Figure 1 molecules-26-01035-f001:**
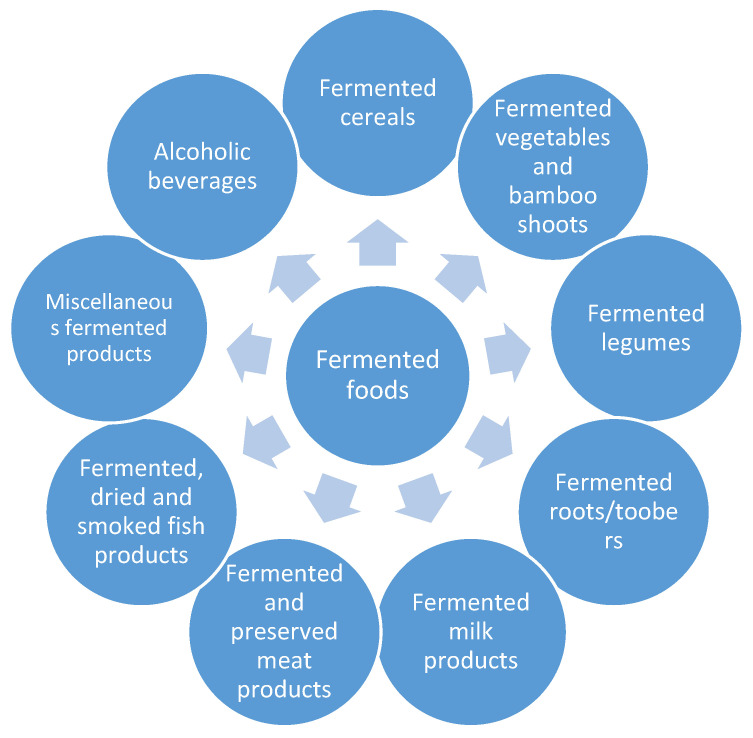
Types of fermented foods [8,9].

**Table 1 molecules-26-01035-t001:** The most common genes related to cold response in *Saccharomyces cerevisiae* yeast [81,82,83,84,85,86].

Name of Gene	Function
*NSR1*	Pre-rRNA processing
*TIP1*	Ribosome biogenesis
*TIR1, TIR2*	Maintenance of cell wall
*INO1, OPI3*	Phospholipid synthesis
*OLE1*	Fatty acid desaturation
*TPS1, TPS2*	Trehalose synthesis
*HSP12, HSP26*	Maintenance of cell morphology, cell adhesion, and germ tube formation
*GPD1*	Glycerol synthesis
*BFR2*	Cell growth through protein trafficking to the Golgi apparatus
*CCTa*, *CCTb*	Cell growth
*ERG10*	Maintenance of cell wall

**Table 2 molecules-26-01035-t002:** Examples of volatile organic compounds present in food products [42,104,130,133,140,141,142].

Compound Name	Odor
**Higher Alcohols**	
Isoamyl alcohol (3-methylbutanol)	Fusel, oil, alcoholic, whiskey, fruity, banana
Amyl alcohol (2-methylbutanol)	Fusel, oil, sweet, balsamic
Isobutyl alcohol (2-methylpropanol)	Ethereal, winey, cortex
Cinnamyl alcohol	Sweet, balsam, hyacinth, spicy, green, powdery, cinnamyl
n-Propyl alcohol (1-propanol)	Alcoholic, fermented, fusel, tequila, musty, yeasty, sweet, fruity, apple, pear
Isopropanol	Alcoholic, musty, woody
n-Butanol	Fusel, oily, sweet, balsamic, whiskey
n-Amyl alcohol	Fusel, oily, sweet, balsamic
2-Phenyloethanol	Rose-honey-like, floral
**Organic Acids**	
Acetic acid	Pungent, acidic, cheesy, vinegar
Propanoic acid	Pungent, acidic, cheesy, vinegar
Butyric acid	Sharp, acetic, cheesy, buttery, fruity
Lactic acid	Sour, acidic
Succinic acid	Sour, acidic
Pyruvic acid	Sharp, sour, acetic, caramellic
Oxalic acid	Bland, mild, caramellic
**Esters**	
Ethyl acetate	Ethereal, fruity, sweet, weedy, green
2-Phenylethyl acetate	Sweet, honey, floral, rose, yeasty, honey, cocoa, balsamic
Isoamyl acetate	Sweet, fruity, banana, solvent
Isobutyl acetate	Sweet, fruity, ethereal, banana, tropical
Butyl butanoate	Sweet, fruity, fresh, diffusive, ripe
Ethyl caproate	Sweet, fruity, pineapple, waxy, green, banana
Hexyl acetate	Fruity, green, apple, banana, sweet
**Phenols**	
4-Ethylguaiacol	Spicy, smoky, bacon, phenolic, clove
4-Vinylphenol	Chemical, phenolic, medicinal, sweet, musty, meaty
Cresol	Smoky, woody, musty/dusty
Carvacrol	Spice, woody, camphor, thymol
2,6-Dimethoxyphenol	Smoky, phenolic, balsamic, bacon, powdery, woody
**Aldehydes and Ketones**	
Acetaldehyde	Pungent, ethereal, aldehydic, fruity
Isobutyrylaldehyde	Banana, melon, varnish, cheese
Isovaleraldehyde	Unripe banana, apple, cheese, amylic
2-Methylbutanal	Green grass, fruity
Octanal	Citrus, orange, green, peely
Nonanal	Rose, fresh, orris, orange, peely
Decanal	Sweet, orange, peely, citrus, floral
Phenylacetaldehyde	Green, sweet, floral, hyacinth, clover, honey, cocoa
Diacetyl	Strong, buttery, sweet, creamy, pungent, caramel
Acetoin	Sweet, buttery, creamy, dairy, milky, fatty
Butanedione	Buttery, sweet, creamy, pungent, caramellic
4-Hydroxyphenyl-2-butanone	Sweet, berry, jammy, raspberry, ripe, floral

**Table 3 molecules-26-01035-t003:** The most common mesophilic yeasts adapted to food processing at low temperatures [143,144,145,146].

Yeast Strain	Fermented Product
*Saccharomyces cerevisiae*	Wine, beer
*Brettanomyces bruxellensis*	Wine, beer
*Saccharomyces pastorianus*	Beer
*Torulaspora delbrueckii*	Wine
*Pichia anomala*	Wine

**Table 4 molecules-26-01035-t004:** Parameters of the fermentation process.

	Yeast Concentration (g/kg Flour)		
	20	40	60
**Fermentation Temperature** (°C)	**Fermentation Time**		
5	21 h	3 h 30 min	3 h
15	3 h	1 h	35 min
35	50 min	25 min	15 min

**Table 5 molecules-26-01035-t005:** Parameters of the fermentation process.

	Yeast Concentration (% of Flour)		
	2	4	6
**Fermentation Temperature** (°C)	**Fermentation Time**		
8	20 h 15 min	8 h 45 min	4 h 50 min
16	7 h 30 min	4 h	3 h
32	2 h	1 h 40 min	1 h 10 min

**Table 6 molecules-26-01035-t006:** The most common mesophilic yeasts adapted to food processing at lower temperatures.

Research	Yeast Strain	Comment
Samoticha and co-workers	*Saccharomyces cerevisiae*	Type of yeast and lower temperatures are key factors for obtaining good sensory profiles of wine. The largest amounts of esters in wine were produced by *S. cerevisiae* at 12 °C.
Kanellaki and co-workers	Immobilized *Saccharomyces cerevisiae* YM-84 and YM-126	Low temperature can potentially lead to better aroma and taste in wine.
Kregiel and co-workers	Immobilized *Saccharomyces cerevisiae* and *Saccharomyces pastorianus*	Low temperature can potentially lead to obtaining a better sensory profile during fermentation.
Torija and co-workers	*Saccharomyces cerevisiae*	Low temperature leads to the formation of more volatile organic compounds (VOCs) and less ethanol in wine.
Molina and co-workers	*Saccharomyces cerevisiae*	At 15 °C, the overall concentration of volatiles was significantly higher than at 28 °C.
Bakoyianis and co-workers	Immobilized *Saccharomyces cerevisiae* (Vinsanto strain)	Wine fermented at low temperature (7 °C) was considered to have improved aroma and taste compared to wine fermented at higher temperatures (up to 27 °C).
Birch and co-workers	*Saccharomyces cerevisiae*	Fermented at 5 °C with a yeast concentration of 60 g/kg flour, bread crumb was found to contain large amounts of esters.
Nor Qhairul Izzreen and co-workers	*Saccharomyces cerevisiae*	Fermentation with a larger amount of yeast at 32 °C gave better results for bread crust, but the formation of key esters increased at 16 °C.

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
