# Peer review of "Yeast Fermentation at Low Temperatures: Adaptation to Changing Environmental Conditions and Formation of Volatile Compounds"

_molecules, 2021, doi:10.3390/molecules26041035_

Round 1
Reviewer 1 Report
The manuscript draft entitled “Yeast fermentation at low temperatures: adaptation to changing environmental conditions and formation of volatile compounds”. This review is written clearly and fluently, but additional discussion and some modifications might be needed before acceptance before publication.
Major
-Section 2.1. might need more information related to the genetic mechanisms of the yeasts adapting low temperatures.
-Line 246. “The VOCs used in the food industry are mostly synthesized by microorganisms”. I don’t think this is entirely true. The VOCs in the food can be “produced” by chemical rxns in food processings, such as maillard rxn, release from non-volatile precursors in foods, and microbial metabolisms.
-Line 341-375. Does it have anything to do with the yeast fermentation at low temperature? If not, the authors are encouraged to put some more relevant references.
-Overall, it is suggested that the authors can add more info and examples regarding the benefits of yeast fermentation at low temperatures and the possible genetic mechanisms for yeast acclimatization.
Minor
-Line 40, Yarrowia lipolytica?
Author Response
Dear Reviewer 1,
We are very grateful for your thorough review of our paper, which has helped us to improve the quality of our publication. We took all your remarks into consideration, and have revised our paper in accordance with your comments. Changed sections have been highlight in yellow. Our point-by-point responses to comments are detailed on the following pages:
- Section 2.1. might need more information related to the genetic mechanisms of the yeasts adapting low temperatures.
We have supplemented introduction in section 2 with information regarding bacteria (line 197-209). We have created section 2.1.1 in which information was also added (line 244-260).
- Line 246. “The VOCs used in the food industry are mostly synthesized by microorganisms”. I don’t think this is entirely true. The VOCs in the food can be “produced” by chemical rxns in food processings, such as maillard rxn, release from non-volatile precursors in foods, and microbial metabolisms.
This sentence was corrected according to the Reviewer’s suggestion (line 353-361).
- Line 341-375. Does it have anything to do with the yeast fermentation at low temperature? If not, the authors are encouraged to put some more relevant references.
From the perspective of our research, these yeasts have the potential to ferment at the low temperature. As a response to the Reviewer’s remark, we have created a new section 3.2. “A brief study of alternative yeasts used in bread-making” (line 578).
- Line 40, Yarrowia lipolytica?
Corrected.
- Overall, it is suggested that the authors can add more info and examples regarding the benefits of yeast fermentation at low temperatures and the possible genetic mechanisms for yeast acclimatization.
We have improved our manuscript with additional information and examples (lines 442-448, 477-486, 487-501).Moreover, we have put a Table with the most important genes responsible for yeast acclimation to low temperatures (line 260).
We hope you will find our article after correction suitable for publication in Molecules.
Yours faithfully,
Joanna Berłowska & Wiktoria Liszkowska
Reviewer 2 Report
The manuscript “Yeast fermentation at low temperatures: adaptation to changing environmental conditions and formation of volatile compounds” discusses the importance of yeasts in biotechnological processes, trying to emphasize the adaptation of yeast to low-temperature processes and their applications in the food industry. Also, the authors discuss the main organic volatile compounds from food products and describe the basic methods of characterizing volatile metabolites.
However, despite the authors’ efforts, the paper does not discuss critically the information, and the content is not well selected and enough comprehensive for the chosen topic and the stated purpose.
The paragraph “Conducting biotechnological processes at low temperature” must be rewritten by targeting more specific information and examples of yeast strains, at least.
The paragraph “Sensory profile: impact of volatile organic compounds on the final product” is discussed poorly and unrelated to the paper’s purpose.
Moreover, paragraph 2.2. regarding the fermentation of lager beer is included in the second paragraph as an example of low-temperature fermentation, even if discussing some of the volatile compounds. Nevertheless, the other paragraphs Winemaking at low temperature and Bread-making at low temperature are included in the 3rd part. The authors must be consistent in discussing these applications. Moreover, the volatile compounds in the case of wine-making at low temperature are presented only in general terms, even if related to a paragraph that it’s supposed to analyze in detail these aspects.
Paragraph 3.2. Bread-making at low temperature
- lines 327-340- what kind of yeast? It is very important information to mention if is normal bakery yeast or not, strain type, etc
-also, the fermentation time is another important parameter of fermentation, with a crucial influence on VOS and it is not mentioned
It is not clear why aspects of the quantification methods for volatile organic compounds should be discussed in this study?
Author Response
Dear Reviewer 2,
We are very grateful for your thorough review of our paper, which has helped us to improve the quality of our publication. We took all your remarks into consideration, and have revised our paper in accordance with your comments. Changed sections have been highlight in yellow. Our point-by-point responses to comments are detailed on the following pages:
- The paragraph “Conducting biotechnological processes at low temperature”must be rewritten by targeting more specific information and examples of yeast strains, at least.
According to the Reviewer’s suggestion, we have made a necessary changes in the paragraph “Conducting biotechnological processes at low temperature”. We have supplemented this section with information about genes related to the low temperature adaptation (line 260) and have created a new short section related to the psychrophilic, psychrotrophic and cryotolerant yeasts (line 214).
- The paragraph “Sensory profile: impact of volatile organic compounds on the final product” is discussed poorly and unrelated to the paper’s purpose.
The paragraph was supplemented with more specific information about VOCs formation (lines 367-439) and Table 2 was updated with more specific compounds (line 443). Moreover, we have added a few new research to this paragraph (line 442-448, 477-484, 487-501).
- Moreover, paragraph 2.2. regarding the fermentation of lager beer is included in the second paragraph as an example of low-temperature fermentation, even if discussing some of the volatile compounds. Nevertheless, the other paragraphs Winemaking at low temperature and Bread-making at low temperature are included in the 3rd
This paragraph has been moved to the section 3.1. Food processing conducted with yeast at low temperatures as an example of low-temperature process (lines 449-469). All examples of the low-temperature fermentation are then in the one section.
- Paragraph 3.2. Bread-making at low temperature (We have merged two paragraphs in the one 1. “Food processing conducted with yeast at low temperatures”)
- lines 327-340- what kind of yeast? It is very important information to mention if is normal bakery yeast or not, strain type, etc.
Corrected.
-also, the fermentation time is another important parameter of fermentation, with a crucial influence on VOS and it is not mentioned
We have supplemented these studies with tables containing concentration of yeast, temperature and time of the fermentation process (Table 3 and Table 4).
- It is not clear why aspects of the quantification methods for volatile organic compounds should be discussed in this study?
We have decided to remove this paragraph from the manuscript.
We hope that you will find our article after correction suitable for publication in Molecules.
Yours faithfully,
Joanna Berłowska & Wiktoria Liszkowska
Reviewer 3 Report
General comment:
With their review article, the two authors want to inform the reader about the relation between the adaptation of industrial yeasts to low temperatures and the formation of volatile aroma compounds. Unfortunately, the manuscript does not quite live up to this claim. Instead, it begins by describing areas of application of yeasts in biotechnology that have already been described many times. The aroma compounds produced by yeasts listed in the tables are also well known from reference books. The last chapter on the detection of volatile compounds seems like a foreign body. In addition, it receives misleading statements: SPME is not an alternative to GC-MS but an upstream technique.
So, overall, there is quite an imbalance in the study between intention and facts. I recommend supplementing the manuscript with a listing of cryotolerant yeasts and, most importantly, providing more information on the cold-related genes involved. Again, tables on genomic and physiological adaptations would be useful.
Overall, however, the intention of the study is quite interesting and well written. After making the necessary additions and corrections, the manuscript can be resubmitted and reviewed.
Special comments:
Line
20: fermentation processes were….
123: Saccharomyces cerevisiae strains
126: S. cerevisiae yeasts
131: at least the most important genes should be listed
245: Table 2 the subtitle Aldehydes appears twice
258: reason that fermented..
263: Small-chain fatty acids are produced?
361: The passage with Kluyveromyces: what is the relation to cold-adaptation?
References: species names should be also in italics here; check for uniform style; letters are arbitrary sometimes in small or in large capitals
Author Response
Dear Reviewer 3,
We are very grateful for your thorough review of our paper, which has helped us to improve the quality of our publication. We took all your remarks into consideration, and have revised our paper in accordance with your comments. Changed sections have been highlight in yellow. Our point-by-point responses to comments are detailed on the following pages:
- Instead, it begins by describing areas of application of yeasts in biotechnology that have already been described many times. The aroma compounds produced by yeasts listed in the tables are also well known from reference books.
This part of the Introduction has been shortened and rewritten. The application of yeast in the food industry has been specified and Figure 1 has been added to summarize types of microbial-derived fermented products (lines 36-37). Table with aroma compounds also have been supplemented with additional compounds (line 440).
- The last chapter on the detection of volatile compounds seems like a foreign body. In addition, it receives misleading statements: SPME is not an alternative to GC-MS but an upstream technique.
We have decided to remove this paragraph from the manuscript.
- I recommend supplementing the manuscript with a listing of cryotolerant yeasts and, most importantly, providing more information on the cold-related genes involved. Again, tables on genomic and physiological adaptations would be useful.
We have created a new short section related to the psychrophilic, psychrotrophic and cryotolerant yeasts (Section 2.1.) (line 214).. Genes responsible for adaptation to the low temperature are listed in the Table 1 (line 260).
- line 20: fermentation processes were…
Corrected.
- line 123: Saccharomyces cerevisiae strains
Corrected.
- line126: S. cerevisiae yeasts
Corrected.
- line131: at least the most important genes should be listed
Genes are listed in the Table 1 (line 260).
- line245: Table 2 the subtitle Aldehydes appears twice
Corrected.
- line258: reason that fermented..
Corrected.
- line263: Small-chain fatty acids are produced?
Corrected.
- line 361: The passage with Kluyveromyces: what is the relation to cold-adaptation
From the perspective of our research, these yeasts have the potential to ferment at the low temperature. As a response to the Reviewer’s remarks, we have created a new section 3.2. “A brief study of alternative yeasts used in bread-making” (line 578).
- References: species names should be also in italics here; check for uniform style; letters are arbitrary sometimes in small or in large capitals
Corrected.
We hope that you will find our article after correction suitable for publication in Molecules.
Yours faithfully,
Joanna Berłowska & Wiktoria Liszkowska
Round 2
Reviewer 1 Report
This manuscript should be good for publication.
Author Response
We are very grateful for your thorough review of our paper, which has helped us to improve the quality of our publication. Thank you.Reviewer 2 Report
I agree with this form of the manuscript
Author Response
We are very grateful for your thorough review of our paper, which has helped us to improve the quality of our publication. Thank you.
Reviewer 3 Report
The authors have carefully followed up on all of the reviewer's suggestions for improvement. In the revised form, the excellent manuscript can be published without any further objections.
Author Response

(The authors gave the same response as above.)
